# Serum Anti-Mullerian Hormone Levels Might Indicate Premenopausal Endometrial Lesions

**DOI:** 10.3390/diagnostics13213301

**Published:** 2023-10-25

**Authors:** Yingsha Yao, Liujing Shi, Xiaoming Zhu

**Affiliations:** 1Women’s Hospital, School of Medicine, Zhejiang University, Hangzhou 310006, China; yaoyingsha@zju.edu.cn (Y.Y.); 22018460@zju.edu.cn (L.S.); 2Key Laboratory of Women’s Reproductive Health of Zhejiang Province, Women’s Hospital, School of Medicine, Zhejiang University, Hangzhou 310006, China

**Keywords:** polycystic ovary syndrome, endometrial proliferative lesions, anti-Mullerian hormone, threshold value, regression model

## Abstract

Background: Endometrial proliferative lesions (EPL) usually refer to endometrial hyperplasia (EH) and endometrial cancer (EC). Among patients with premenopausal EPL who wish to preserve their fertility, only those with EH and early-stage EC have the possibility to undergo fertility preservation therapy. However, there is currently a lack of specific and reliable screening criteria and models for identifying these patients. Methods: This study utilized a retrospective diagnostic study design. The training set included medical record information that met the criteria between August 2017 and October 2022, while the validation set consisted of medical record information that met the criteria from November 2022 to May 2023. The endometrial pathological test served as the gold standard. The serum anti-Mullerian hormone (AMH) level before endometrial sampling and a regression model were employed to predict EPL. Results: The study included a total of 1209 patients with PCOS (1119 in the control group and 90 in the endometrial proliferative lesion group) and 5366 women without PCOS (5249 in the control group and 117 in the proliferative lesion group). In the case of PCOS patients aged 20–39 years, the most effective screening threshold for AMH was found to be a serum AMH level of ≤5.39 ng/mL. The model used for this group was logit(p) = −2.562 − 0.430 × AMH + 0.127 × BMI + 1.512 × hypertension + 0.956 × diabetes −1.145 × regular menstruation. On the other hand, for non-PCOS women aged 20–39 years, the optimal screening threshold for AMH was determined to be a serum AMH value of ≤2.18 ng/mL. The model used for this group was logit(p) = −3.778 − 0.823 × AMH + 0.176 × BMI + 2.660 × diabetes −1.527 × regular menstruation −1.117 × dysmenorrhea. It is important to note that all of these findings have successfully passed internal verification. Conclusion: For PCOS and non-PCOS women aged 20–39 years, the serum AMH test and related multiple regression models were obtained for the warning of EPL.

## 1. Introduction

Endometrial proliferative lesions (EPL) usually refer to endometrial hyperplasia (EH) and endometrial cancer (EC). EC is the sixth most common cancer worldwide for females, and therefore represents a significant public health related issue [1,2]. In 1983, Bokhman proposed a binary classification of the pathogenesis of EC: type I EC (endometrioid adenocarcinoma) and type II EC (non-endometrioid cancer, namely other types of EC) [3], among which type I EC mostly had EH as the background. Without intervention, the risk of atypical endometrial hyperplasia (AEH) developing into EC was as high as about 30% [4,5]. The incidence and associated mortality of EC are rising in the world, either in the transitioning or transitioned countries [1,2,6,7]. Although the vast majority of EC occurs after menopause, up to 14% of patients are premenopausal, of which 4% occur before the age of 40 [8]. With the increase in the obesity rate and the decline in the fertility rate, the proportion of EC before menopause has increased significantly, and the proportion before 40 years old is rising [9].

The occurrence of EPL can have an impact on the reproductive outcome of premenopausal patients. However, only patients diagnosed with EH and early low-stage EC have the chance to undergo a treatment plan for fertility preservation. It is crucial for premenopausal patients to detect EPL early and preserve their reproductive function through progesterone therapy [10].

The preferred way to screen for the disease is non-invasive or minimally invasive. Current research has primarily concentrated on detecting EPL in patients with symptoms, particularly abnormal uterine bleeding [11,12]. However, in premenopausal women, the association between abnormal uterine bleeding and EPL is significantly stronger compared to the association between abnormal uterine bleeding after menopause and EPL [13]. Many researchers have focused on biomarkers that can predict EPL. A biomarker is defined by the US National Cancer Institute as “a biological molecule found in blood, other body fluids, or tissues that is a marker of a normal or abnormal process or disease” [14]. The World Health Organization defines a biomarker as “any substance, structure, or process that can be measured in the human body or its products and influence or predict the outcome or incidence of disease” [15]. 

The sampling process based on blood is convenient, easy to operate, and less traumatic. It is often used as an indicator for disease screening and follow-up after treatment [16,17,18]. Other EPL-related biomarkers, such as gene-related markers mRNA, are still under development [19,20,21,22]. Currently, there are no effective and cost-effective biomarkers for the diagnosis of EPL. The application of mature biomarkers to other diseases may provide new ideas for clinical research.

Anti-Mullerian hormone (AMH), also known as Mullerian anti-substance, is mainly produced by secondary, presinus and early sinus follicles (less than 8 mm in diameter) [23]. AMH is a well-established biomarker commonly utilized to assess ovarian reserve and guide assisted reproductive strategies [24,25]. However, its primary application lies within the realm of reproductive endocrinology. 

This paper intends to use AMH as a candidate biomarker for screening EPL, for providing a new reference index for screening EPL and reducing unnecessary invasive procedures (making invasive examination more targeted). 

Polycystic ovary syndrome (PCOS) is a commonly occurring reproductive endocrine metabolic disease. Research has shown that the level of serum AMH in PCOS patients is 2–4 times higher than that of healthy women [26,27,28], and this trend is observed across different types of PCOS patients [29]. Serum AMH levels peak after sexual function has matured (around 20 years of age), gradually decline with age, and become undetectable after menopause [23]. Based on this, the study focused on women aged 20–39 years and classified them into PCOS and non-PCOS groups.

## 2. Methods

This study utilized a retrospective diagnostic study design. The study was approved by the Ethical Review Committee for clinical research (Approval No. IRB-20210312-R). 

### 2.1. Inclusion Criteria and Exclusion Criteria

The study focused on premenopausal women with PCOS and those without PCOS. These women underwent a pathological examination of the endometrial tissue after a serum AMH test. The training set consisted of medical record information meeting the criteria from August 2017 to October 2022, while the validation set comprised medical record information meeting the criteria from November 2022 to May 2023. 

The subjects were required to have recent AMH test reports, conducted within three months prior to surgery. The AMH levels considered in this study were based on the laboratory test results of our hospital, accounting for any variations in AMH results due to different detection platforms or methods [30,31]. The pathological diagnosis report of endometrial samples is the gold standard for EPL. 

Current international evidence-based guidelines [32] recommend the use of the 2003 Rotterdam criteria recommended by the European Society of Human Reproduction and Embryology (ESHRE) and the American Society for Reproductive Medicine (ASRM) [33], which is one of the widely used diagnostic criteria for PCOS. Two of the following three items exist: (1) ovulatory dysfunction (OD); (2) hyperandrogenism (HA); (3) polycystic ovarian morphology (PCOM). Other causes of hyperandrogenemia are excluded (e.g., hyperprolactinemia, thyroid disease, congenital adrenal hyperplasia, Cushing’s syndrome, etc.).

The exclusion criteria for this study include incomplete medical record data, a clear history of endocrine tumors, definite abnormalities in chromosome number or structure, or a genetic disease related to ovarian function. Patients with other conditions that could clearly affect serum AMH levels, such as those with pathological or physiological ovarian dysfunction, were excluded from the study. 

The diagnosis of ovarian insufficiency was made based on the Bologna criteria [34] for poor ovarian response (POR). One of the following three items exist: (1) advanced maternal age (≥40 years) or any other risk factors for ovarian hypofunction/insufficiency; (2) previous POR, i.e., ≤3 oocytes, conventional stimulation program; (3) an abnormal ovarian reserve test, i.e., Antral Follicle Count (AFC) < 5–7 follicles or AMH < 0.5–1.1 ng/mL [34].

Additionally, participants with severe diseases of important organs such as cardiovascular, lung, liver, kidney, or nerve diseases are also excluded.

### 2.2. Sample Estimation

In this study, a small sample pre-test was conducted to determine the sensitivity and specificity. The sample size was calculated using the formula $n=\frac{U_{\alpha /2}^{2}*p(1-p)}{\delta^{2}}$, where α was set to 0.05 and unilateral Uα/2 was 1.96. The sensitivity and specificity values were used to estimate the sample size for both the control group and the lesion group, with an allowable error of δ of 0.1.

### 2.3. Data Acquisition

The study participants were assessed for several basic characteristics including age, education level, menstrual history (age of menarche, menstrual cycle length, duration of menstrual period, and history of dysmenorrhea), physique (height, weight, and signs of hirsutism/acne), reproductive history (number of pregnancies and number of births), and family history of tumors as well as smoking history.

PCOS phenotype [35]: Type A was characterized by hyperandrogenism (HA), ovulatory dysfunction (OD) and Polycystic ovarian morphology (PCOM). Type B has HA and OD without PCOM; Type C has HA, PCOM but no OD; Type D has OD, PCOM but no HA.

Previous underlying diseases were considered: common chronic diseases such as hypertension and diabetes, a history of thyroid and adrenal endocrine abnormalities, a history of liver and kidney metabolism abnormalities, a history of surgery, radiotherapy, and chemotherapy for neoplastic diseases, a history of chromosome-related diseases and gene-related diseases, and a history of drug use including but not limited to long-term progesterone use or periodic contraceptive drugs.

Serological indicators were examined in the 3 months prior to endometrial biopsy. These indicators included serum AMH, basal levels of follicular stimulating hormone (bFSH), luteinizing hormone (bLH), estradiol (bE2), and progesterone (bP). Additionally, other endocrine levels such as TT and prolactin (PRL), as well as tumor markers including carcinoma embryonic antigen (CEA), alpha-fetoprotein (AFP), CA-125, CA-153, and CA-199 were analyzed.

Serum AMH levels were measured by electrochemical luminescence (Roche Diagnostics, Mannheim, Germany), The lower limit and upper limit of AMH levels were 0.01 ng/mL and 24 ng/mL, respectively. The intra-assay and inter-assay coefficients of variation were less than 10%.

When recording serological hormone indices, if they were lower than the lower limit of detection, the lower limit of detection was recorded. The same index was tested repeatedly within three months before surgery, and the average value was taken. In the short term, the results of multiple examinations were not uniform, and the patients with a high lesion degree were selected.

### 2.4. Statistical and Analytical Methods

The data were input using Microsoft Excel (Microsoft Corporation, Redmond, WA, USA) and double-checked. 

Statistical analysis of all data was performed using the statistical software SPSS 26.0. After normalizing and transposing the AMH data, any values outside the range of (−3, +3) were considered outlier values and were removed. The multiple regression analysis was performed to obtain a regression model (the input method was adopted, as well as the step-forward method).

MedCalc (MedCalc Software Ltd., Ostend, Belgium) was utilized for calculating the optimal critical value of AMH warning EPL. The software was also employed to generate the receiver operating characteristic (ROC) curve for AMH warning threshold and regression model warning EPL, and subsequently, the warning efficiency was calculated and compared. The warning efficiency indicators considered were the area under the curve (AUC) of the ROC curve, sensitivity, specificity, Youden index, negative likelihood ratio, and positive likelihood ratio. An ideal AUC would be close to 1 and at least 0.7, indicating clinical utility as a biomarker [36].

## 3. Results

### 3.1. Estimate the Sample Size

In the pre-experiment, the sensitivity and specificity of the test for PCOS patients were approximately 0.7 and 0.75, respectively. The calculated sample size for the EPL group should be no less than 89 individuals, while the control group should have a sample size of at least 79 people. 

Similarly, for non-PCOS patients, the sensitivity and specificity of the pre-experiment were about 0.7 and 0.8, respectively. The calculated sample size for the EPL group should be no less than 89 individuals, and the control group should have a sample size of at least 67 people.

### 3.2. Baseline Information

Clinical data were collected retrospectively from premenopausal women with PCOS and non-PCOS who underwent serum AMH testing and received a pathological diagnosis of endometrial tissue between August 2017 and October 2022 (Figure 1). The study included a total of 1209 patients with PCOS (1119 in the control group and 90 in the endometrial proliferative lesion group) and 5366 women without PCOS (5249 in the control group and 117 in the proliferative lesion group). Table 1 presents the basic information, relevant test results, and differences between the female PCOS control group and the proliferative lesion group. Similarly, Table 2 displays the basic information, relevant test results, and differences between the non-PCOS female control group and the proliferative lesion group.

### 3.3. AMH Optimal Threshold and Warning Model for PCOS

As shown in Table 1, there were statistically significant differences (*p* < 0.05) between the control group and PCOS women with EPL in several basic characteristics (pregnancy time, body mass index (BMI), phenotypic composition/structure of PCOS, prevalence of hypertension and diabetes, menstrual regularity, dysmenorrhea history) and blood test results (AMH, bE2, bP). The input methods that had significant differences were used for binary logistic regression analysis, and the specific results of the logistic regression are presented in Table 3, based on the multi-factor binary logistic regression model logit(p) = −2.562 − 0.430 × AMH + 0.127 × BMI + 1.512 × hypertension + 0.956 × diabetes − 1.145 × regular menstruation. The optional threshold value of AMH for predicting EPL in PCOS patients is 5.39 ng/mL (less than or equal to this threshold suggests the patients may have EPL).

The ROC curves of the AMH threshold value and regression model for screening EPL were presented in Figure 2. The corresponding AUC, sensitivity, specificity, Youden index, positive likelihood ratio, and negative likelihood ratio were calculated. Specific data can be found in Table 4.

The AUC, sensitivity, specificity, Youden index, positive likelihood ratio, and negative likelihood ratio of the AMH threshold value were as follows: AUC = 0.788, sensitivity = 61.11%, specificity = 86.95%, Youden index = 0.4806, positive likelihood ratio = 4.68, and negative likelihood ratio = 0.45. Similarly, for the AMH threshold value, the AUC, sensitivity, specificity, Youden index, positive likelihood ratio, and negative likelihood ratio were as follows: AUC = 0.833, sensitivity = 73.26%, specificity = 79.61%, Youden index = 0.5287, positive likelihood ratio = 3.59, and negative likelihood ratio = 0.34. Based on the AUC values, both serum AMH testing and regression models showed moderate effectiveness in predicting EPL in PCOS patients aged 20 to 39 years. However, the regression model was found to be superior to the serum AMH test (*p* < 0.05).

The verification set consisted of 153 samples, with 137 in the control group and 16 in the EPL group. There were no significant differences in baseline information and blood test results between the training set and verification set, as shown in Appendix A. Among the validation sets, the sensitivity and specificity of the AMH recommended threshold (≤5.39 ng/mL) for predicting EPL in PCOS patients were 62.50% and 79.56%, respectively, with an overall prediction accuracy of 77.78%. The sensitivity and specificity of the regression model for predicting EPL in PCOS patients were 80.00% and 77.94%, respectively, with an overall prediction accuracy of 78.43% (Table 5).

### 3.4. AMH Optimal Threshold and Warning Model for Non-PCOS

As shown in Table 2, there were statistically significant differences (*p* < 0.05) between the control group and non-PCOS women with EPL in several basic characteristics (age, birth time, BMI, hypertension and diabetes, menstrual regularity, dysmenorrhea history) and blood test results (AMH, bE2, bP, CEA). The input methods that had significant differences were used for binary logistic regression analysis, and the specific results of the logistic regression are presented in Table 6, based on the multi-factor binary logistic regression model: logit(p) = −3.778 − 0.823 × AMH + 0.176 × BMI + 2.660 × diabetes −1.527 × regular menstruation −1.117 × dysmenorrhea. The optional threshold value of AMH for predicting EPL in non-PCOS patients is 2.18 ng/mL (less than or equal to this threshold suggests that patients may have EPL).

The ROC curves of the AMH threshold value and regression model for screening EPL were presented in Figure 3. The corresponding AUC, sensitivity, specificity, Youden index, positive likelihood ratio, and negative likelihood ratio were calculated. Specific data can be found in Table 7.

The AUC, sensitivity, specificity, Youden index, positive likelihood ratio, and negative likelihood ratio of the AMH threshold value were as follows: AUC = 0.810, sensitivity = 71.79%, specificity = 77.31%, Youden index = 0.4910, positive likelihood ratio = 3.16, and negative likelihood ratio = 0.36. Similarly, for the AMH threshold value, the AUC, sensitivity, specificity, Youden index, positive likelihood ratio, and negative likelihood ratio were as follows: AUC = 0.847, sensitivity = 74.77%, specificity = 82.04%, Youden index = 0.5681, positive likelihood ratio = 4.16, and negative likelihood ratio = 0.31. Based on the AUC values, both serum AMH testing and regression models showed moderate effectiveness in predicting EPL in PCOS patients aged 20 to 39 years. However, the regression model was found to be superior to the serum AMH test (*p* < 0.05).

The verification set consisted of 736 samples, with 716 in the control group and 20 in the EPL group. There were no significant differences in baseline information and blood test results between the training set and verification set, as shown in Appendix A. Among the validation sets, the sensitivity and specificity of the AMH recommended threshold (≤2.18 ng/mL) for predicting EPL in non-PCOS women were 85.00% and 78.63%, respectively, with an overall prediction accuracy of 78.80%. The sensitivity and specificity of the regression model for predicting EPL in non-PCOS women were 88.89% and 83.82%, respectively, with an overall prediction accuracy of 83.97% (Table 8).

## 4. Discussion

In this study, we found that the serum AMH median level in PCOS patients was more than twice that of non-PCOS women, which is consistent with previous studies [26,27,28]. The study revealed a significantly higher prevalence of EPL in PCOS patients compared to non-PCOS women with normal reproductive function (7.44% vs. 2.18%, *p* < 0.05), aligning with previous research findings [37]. The results of this single-center retrospective study suggest that the patients included were somewhat representative, and the data obtained have some reference value. The study found that serum AMH can serve as an indicator for EPL in women aged 20–39 years, both in those with PCOS and those without.

Currently, the evaluation of EPL in women of reproductive age, including both PCOS patients and non-PCOS women, primarily involves ultrasound examination and cytology examination. Ultrasonic examination assesses the possibility of EPL based on the endometrial thickness and Doppler signal of blood flow [38]. However, it is important to note that the ultrasonic signals of the endometrium in premenopausal women may undergo physiological changes during different stages of the menstrual cycle, which can potentially impact the accuracy of the ultrasonic results for detecting EPL [39,40]. In order to minimize the disadvantages of invasive and traumatic diagnosis of EPL, researchers have been investigating ways to obtain intuitive evidence (pathology) for diagnosing the disease while reducing discomfort. Cytological examination has been successfully used for screening cervical cancer by analyzing cervical shedding cells [41]. However, this method is limited by the sampling tools and uterine morphology, resulting in inadequate sampling satisfaction [42,43,44].

Biomarkers, such as tumor markers, are commonly used in physical examinations to screen for specific cancers [14,19,45]. CA-125 has been found to be useful in predicting tumor metastasis before EC, recurrence, and treatment prognosis [46,47,48,49]. However, the ability of CA-125 to diagnose EPL is still uncertain, and different studies have provided conflicting results. A study conducted by Lina Zhou’s team on patients already diagnosed with AEH suggested that preoperative CA-125 levels could partially distinguish between AEH and EC [18]. On the other hand, a prospective study by Tatiana Cuesta Guardiola’s team found that serum CA-125 was not effective in diagnosing EC [50]. When CA-125 levels are abnormally elevated, caution should be exercised in interpreting its clinical significance in judging EPL [45]. Serum human epididymal protein 4 (HE4) is currently licensed for the diagnosis and monitoring of ovarian cancer recurrence. Since both ovarian and endometrial tissues are derived from Muller’s ducts, these two primary cancers share similarities in etiology, gene expression profile, tumorigenic mechanism, pathological changes, and metastasis characteristics [51,52,53]. Compared to serum CA125, serum HE4 has shown superior specificity in diagnosing EC, but its sensitivity is poor, and it is not suitable for screening for EPL [54,55].

The current lack of reliable biomarkers for screening EPL is addressed in this study. For the first time, the serum AMH test was used to screen for gynecological oncology. The study introduces a warning threshold value for the serum AMH test to predict EPL and proposes a warning model based on AMH. These findings have promising clinical reference value and potential for further research and application, especially in cases where patients are resistant to additional invasive examinations. The serum AMH test and its related regression models serve as a reminder of the risk of EPL, supporting the need for further invasive examinations such as hysteroscopy. Moreover, since AMH itself is already an established detection method, its application in new areas is more feasible compared to developing a completely new biomarker.

This study has some limitations. Firstly, the optimal threshold of serum AMH for early warning of EPL and the establishment and verification of regression models are based on single-center data, which limits their representability. Therefore, caution should be exercised in their popularization and use. Secondly, the sample size in this study is limited, and the age range of the population is relatively large, spanning nearly 20 years. To find a more reliable, accurate, and practical serum AMH threshold value, as well as to establish a more efficient regression model for EPL in women of different age ranges, it is necessary to conduct a prospective multi-center large-sample diagnostic study involving a larger population of women.

## 5. Conclusions

This study is the first to introduce AMH into the field of gynecological tumors, aiming to utilize it for EPL. The regression model based on serum AMH detection or the regression model proposed in this study showed moderate effectiveness in early warning of EPL for PCOS patients aged 20 to 39 years and non-PCOS women. However, due to the limitation of data being sourced from a single center, the early warning ability of serum AMH for EPL needs to be further validated through larger multi-center studies.

## Figures and Tables

**Figure 1 diagnostics-13-03301-f001:**
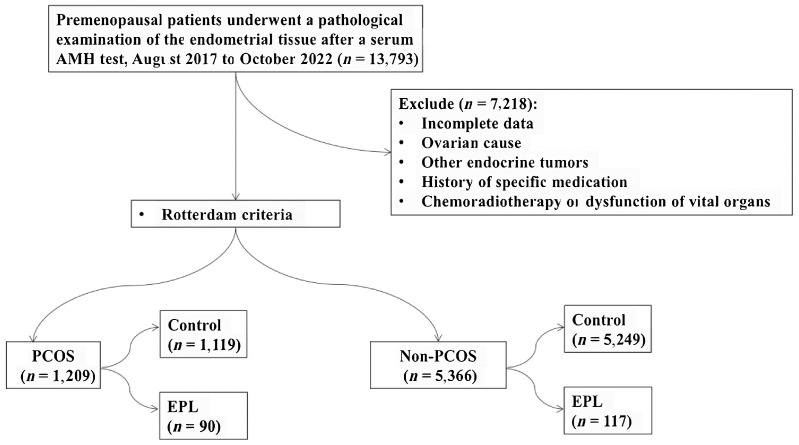
Flow chart of inclusion and exclusion is presented (PCOS: polycystic ovary syndrome; EPL: endometrial proliferative lesions).

**Figure 2 diagnostics-13-03301-f002:**
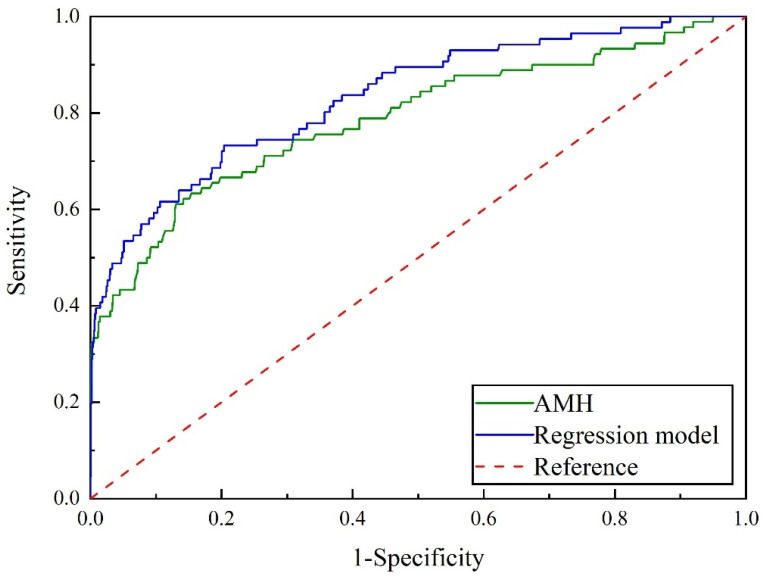
ROC curves correspond to AMH and regression model for PCOS (AMH: anti-Mullerian hormone; PCOS: polycystic ovary syndrome; ROC: receiver operating characteristic).

**Figure 3 diagnostics-13-03301-f003:**
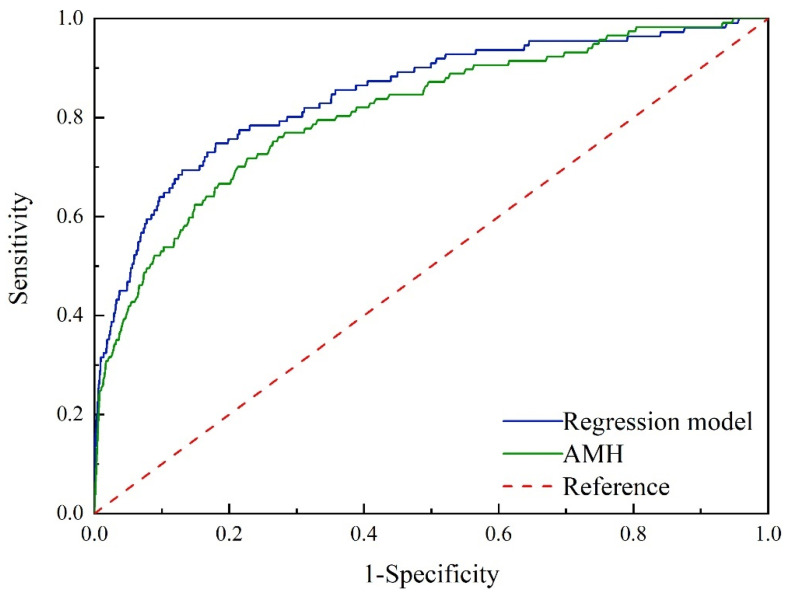
ROC curves correspond to AMH and regression model for non-PCOS (AMH: anti-Mullerian hormone; PCOS: polycystic ovary syndrome; ROC: receiver operating characteristic).

**Table 1 diagnostics-13-03301-t001:** Clinical characteristics of polycystic ovary syndrome are presented.

Feature	Control (*n* = 1119)	EPL (*n* = 90)	*p* Value
Age (year)	30 (28–33)	30.5 (29–35)	0.067
Menophania (year)	13 (13–14)	13 (13–14)	0.417
Menstrual period (day)	6 (5–7)	7 (5–7)	0.176
Gravidity	1 (0–2)	0 (0–1)	<0.001
Parity	0 (0–0)	0 (0–0)	0.999
BMI (kg/m^2^)	22.27 (20.32–24.54)	25.33 (22.17–29.10)	<0.001
PCOS phenotype			
A	376 (33.6%)	23 (25.6%)	0.004
B	244 (21.8%)	19 (21.1%)	
C	112 (10.0%)	2 (2.2%)	
D	387 (35.6%)	46 (51.1%)	
Underlying disease			
Hypertension	23 (2.1%)	15 (16.7%)	<0.001
Diabetes mellitus	24 (2.1%)	10 (11.1%)	<0.001
Higher education	706 (63.1%)	54 (60.0%)	0.559
Menstrual regularity	374 (33.4%)	11 (12.2)	<0.001
Dysmenorrhea	267 (23.9%)	12 (13.3%)	0.032
History of sex hormone use	65 (5.8%)	7 (7.8%)	0.122
Family history of cancer	9 (0.8%)	0 (0.0%)	0.448
Smoking history			0.828
AMH (ng/mL)	7.99 (6.24–10.42)	5.01 (2.61–7.01)	<0.001
Basal endocrine			
bLH (IU/L)	7.63 (5.25–11.84)	8.17 (5.54–11.78)	0.834
bFSH (IU/L)	5.81 (4.99–6.79)	6.24 (5.28–7.21)	0.129
bE_2_ (pmol/L)	106.60 (74.43–134.00)	126.50 (103.93–150.18)	0.003
bP (nmol/L)	0.95 (0.59–1.34)	0.75 (0.41–1.13)	0.014
TT (nmol/L)	1.10 (0.80–1.50)	1.00 (0.70–1.40)	0.177
PRL (ng/mL)	14.60 (11.00–19.80)	14.20 (10.30–19.50)	0.368
Tumor marker			
CEA (ng/mL)	1.10 (0.70–1.50)	1.20 (0.85–1.90)	0.060
AFP (ng/mL)	2.20 (1.60–3.20)	2.30 (1.50–3.55)	0.766
CA-125 (U/mL)	14.20 (10.10–20.50)	14.95 (11.58–22.13)	0.077
CA-153 (U/mL)	8.10 (5.90–11.70)	8.50 (6.48–11.75)	0.318
CA-199 (U/mL)	8.90 (5.80–13.80)	8.50 (5.10–13.10)	0.313

PCOS: polycystic ovary syndrome; EPL: endometrial proliferative lesions; BMI: Body Mass Index; AMH: anti-Mullerian hormone; bFSH: basal follicular stimulating hormone; bLH: basal Luteinizing hormone; bE2: basal Estradiol; bP: basal progesterone; TT: total testosterone; PRL: prolactin; CEA: carcinoma embryonic antigen; AFP: alpha-fetoprotein.

**Table 2 diagnostics-13-03301-t002:** Clinical characteristics of women without polycystic ovary syndrome are presented.

Feature	Control (*n* = 5249)	EPL (*n* = 117)	*p* Value
Age (year)	31 (28–34)	33 (28–38)	0.001
Menophania (year)	14 (13–14)	13 (13–14)	0.087
Menstrual period (day)	6 (5–7)	6 (5–7)	0.547
Gravidity	1 (0–2)	1 (0–2)	0.071
Parity	0 (0–1)	0 (0–1)	0.019
BMI(kg/m^2^)	21.11 (19.53–23.11)	23.78 (20.81–26.56)	<0.001
Underlying disease			
Hypertension	37 (0.7%)	9 (7.7%)	<0.001
Diabetes mellitus	25 (0.5%)	5 (4.3%)	<0.001
Higher education	3262 (62.1%)	67 (57.3%)	0.282
Menstrual regularity	4706 (89.7%)	76 (65.0%)	<0.001
Dysmenorrhea	1496 (28.5%)	15 (12.8%)	<0.001
History of sex hormone use	297 (5.7%)	6 (5.1%)	0.951
Family history of cancer	52 (0.9%)	0 (0.0%)	0.806
Smoking history			0.545
AMH (ng/mL)	3.28 (2.28–4.58)	1.58 (1.11–2.36)	<0.001
Basal endocrine			
bLH (IU/L)	4.70 (3.56–5.94)	4.20 (2.59–6.17)	0.125
bFSH (IU/L)	6.45 (5.52–7.50)	6.38 (4.83–8.73)	0.890
bE_2_ (pmol/L)	102.80 (75.24–132.40)	125.50 (92.00–147.60)	0.016
bP (mol/L)	1.04 (0.70–1.41)	0.73 (0.47–1.12)	<0.001
TT (nmol/L)	0.70 (0.50–1.00)	0.60 (0.33–0.90)	0.060
PRL (ng/mL)	15.50 (11.70–21.50)	15.90 (11.00–22.00)	0.977
Tumor marker			
CEA (ng/mL)	1.00 (0.70–1.50)	1.20 (0.80–1.80)	0.031
AFP (ng/mL)	2.20 (1.60–3.20)	2.30 (1.50–3.25)	0.513
CA-125 (U/mL)	15.70 (11.50–22.60)	17.20 (13.00–24.05)	0.068
CA-153 (U/mL)	8.30 (6.10–11.80)	9.30 (6.28–13.00)	0.081
CA-199 (U/mL)	9.60 (6.30–15.10)	10.60 (7.11–5.20)	0.207

PCOS: polycystic ovary syndrome; EPL: endometrial proliferative lesions; BMI: Body Mass Index; AMH: anti-Mullerian hormone; bFSH: basal follicular stimulating hormone; bLH: basal Luteinizing hormone; bE2: basal Estradiol; bP: basal progesterone; TT: total testosterone; PRL: prolactin; CEA: carcinoma embryonic antigen; AFP: alpha-fetoprotein.

**Table 3 diagnostics-13-03301-t003:** Logistic regression analysis of differential factors for PCOS is presented.

Feature	Single Factor	*p* Value	Multi-Factor	*p* Value
Gravidity	0.719 (0.583–0.886)	0.002	-	-
BMI (kg/m^2^)	1.249 (1.181–1.321)	<0.001	1.136 (1.064–1.212)	<0.001
PCOS phenotype	Dummy variable processing (Type D for reference)	0.012/ 0.137/ 0.009	-	-
Hypertension	9.530 (4.774–19.025)	<0.001	4.536 (1.917–10.730)	0.001
Diabetes mellitus	5.703 (2.636–12.340)	<0.001	2.609 (1.010–6.739)	0.045
Regularity of menstruation	0.277 (0.146–0.528)	<0.001	0.318 (0.160–0.632)	0.001
Dysmenorrhea	0.491 (0.263–0.915)	0.025	-	-
AMH (ng/mL)	0.608 (0.543–0.681)	<0.001	0.651 (0.578–0.733)	<0.001
bE_2_ (pmol/L)	1.005 (1.000–1.011)	0.058	-	-
bP (nmol/L)	1.088 (0.996–1.187)	0.061	-	-

PCOS: polycystic ovary syndrome; BMI: Body Mass Index; AMH: anti-Mullerian hormone; bE2: basal Estradiol; bP: basal progesterone.

**Table 4 diagnostics-13-03301-t004:** Efficacy of AMH and regression models to screen endometrial proliferative lesions in PCOS patients is presented.

Comparative Indices	AMH	Regression Models
AUC	0.788	0.833 *
AUC CI	0.764–0.811	0.811–0.854
Optimal threshold	≤5.39 ng/mL	-
Sensitivity	61.11	73.26
Specificity	86.95	79.61
Youden Index	0.4806	0.5287
LR+	4.68	3.59
LR−	0.45	0.34

Note: * Comparison of AUC differences in screening tests (AMH vs. regression model), Delong test: Z statistic = 2.066, *p* < 0.05; AMH: anti-Mullerian hormone; PCOS: polycystic ovary syndrome; AUC: area under the curve; CI: confidence interval; LR+: positive likelihood ratio; LR−: negative likelihood ratio.

**Table 5 diagnostics-13-03301-t005:** AMH threshold value and regression model for EPL with PCOS is presented.

Sample	Golden Standard	AMH Threshold		Sensitivity	Accuracy	Regression Model		Sensitivity	Accuracy
		Yes	No	Specificity		Yes	No	Specificity	
Train set (*n* = 1209)	EPL	55	35	61.11	85.03	66	24	73.26	79.16
	Non-EPL	146	973	86.95		228	891	79.61	
Validation set (*n* = 153)	EPL	10	6	62.50	77.78	13	3	80.00	78.43
	Non-EPL	28	109	79.56		30	107	77.94	

AMH: anti-Mullerian hormone; PCOS: polycystic ovary syndrome; EPL: endometrial proliferative lesions.

**Table 6 diagnostics-13-03301-t006:** Logistic regression analysis of differential factors for non-PCOS is presented.

Feature	Single Factor	*p* Value	Multi-Factor	*p* Value
Age (year)	1.086 (1.040–1.134)	<0.001	-	0.177
Parity	1.403 (1.076–1.830)	0.012	-	-
BMI (kg/m^2^)	1.247 (1.187–1.309)	<0.001	1.192 (1.084–1.311)	<0.001
Hypertension	11.739 (5.529–24.925)	<0.001	-	0.151
Diabetes mellitus	9.329 (3.507–24.812)	<0.001	14.297 (2.772–73.743)	0.001
Regularity of menstruation	0.214 (0.145–0.316)	<0.001	0.217 (0.103–0.460)	<0.001
Dysmenorrhea	0.369 (0.214–0.636)	<0.001	0.327 (0.125–0.857)	0.023
AMH (ng/mL)	0.332 (0.263–0.418)	<0.001	0.439 (0.313–0.616)	<0.001
bE_2_ (pmol/L)	1.005 (1.000–1.010)	0.065	-	-
bP (nmol/L)	0.262 (0.131–0.523)	<0.001	-	0.051
CEA (ng/mL)	1.074 (0.986–1.170)	0.102	-	-

PCOS: polycystic ovary syndrome; BMI: Body Mass Index; AMH: anti-Mullerian hormone; bE2: basal Estradiol; bP: basal progesterone; CEA: carcinoma embryonic antigen.

**Table 7 diagnostics-13-03301-t007:** Efficacy of AMH and regression models to screen endometrial proliferative lesions in non-PCOS patients is presented.

Comparative Indices	AMH	Regression Models
AUC	0.810	0.847 *
AUC CI	0.799–0.820	0.837–0.857
Optimal threshold	≤2.18 ng/mL	-
Sensitivity	71.79	74.77
Specificity	77.31	82.04
Youden Index	0.4910	0.5681
LR+	3.16	4.16
LR−	0.36	0.31

Note: * Comparison of AUC differences in screening tests (AMH vs. regression model), Delong test: Z statistic = 3.122, *p* < 0.05; AMH: anti-Mullerian hormone; PCOS: polycystic ovary syndrome; AUC: area under the curve; CI: confidence interval; LR+: positive likelihood ratio; LR−: negative likelihood ratio.

**Table 8 diagnostics-13-03301-t008:** AMH threshold value and regression model for EPL without PCOS is presented.

Sample	Golden Standard	AMH Threshold		Sensitivity	Accuracy	Regression Model		Sensitivity	Accuracy
		Yes	No	Specificity		Yes	No	Specificity	
Train set (*n* = 5366)	EPL	84	33	71.79	77.19	87	30	74.77	81.87
	Non-EPL	1191	4058	77.31		943	4306	82.04	
Validation set (*n* = 736)	EPL	17	3	85.00	78.80	18	2	88.89	83.97
	Non-EPL	153	563	78.63		116	600	83.82	

AMH: anti-Mullerian hormone; PCOS: polycystic ovary syndrome; EPL: endometrial proliferative lesions.

## Data Availability

The data sets generated for this study are available on request to the corresponding author.

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
