# Peer review of "Serum Anti-Mullerian Hormone Levels Might Indicate Premenopausal Endometrial Lesions"

_diagnostics, 2023, doi:10.3390/diagnostics13213301_

Round 1

Reviewer 1 Report

The paper introduced AMH as a potential biomarker for endometrial proliferative lesions (EPL). Though the results did not show high sensitivity or specificity, it provided a warning model for EPL based on AMH levels.

The study focused on the age group of 20-39 years of women with PCOS and without PCOS. Is there any data about the population with more advanced age when EPL happens more frequently?

Is there any study combining AMH with other biomarkers?

Author Response

First, thank you very much for your review comments. Regarding your questions, we provide the following feedback:

Due to the influence of increasing age on anti-mullerian hormone (AMH), in order to reduce the influence of age, this study referred to the Bologna criteria for poor ovarian response (POR), and did not study women over 40 years old. In the future, the research team will attempt to explore the ability of serum AMH testing to screen for endometrial proliferative lesions (EPL) in such a subset of the population. In future research, we plan to explore the potential of serum AMH testing to screen for EPL specifically in this age group.

In this study, we included various biomarkers such as serum basal levels of follicular stimulating hormone (bFSH), luteinizing hormone (bLH), estradiol (bE2), progesterone (bP). We also analyzed other endocrine levels such as total testosterone (TT) and prolactin (PRL), as well as tumor markers including carcinoma embryonic antigen (CEA), alpha-fetoprotein (AFP), CA-125, CA-153, and CA-199. To further improve the early warning ability of serum AMH for EPL, a regression model was established. However, the results of the regression analysis indicated that the above biomarkers were not suitable for inclusion in the final regression equation. In future studies, we will continue to explore the possibility of combining AMH with other commonly used biomarkers for early warning of EPL.

Thank you again for taking time out of your busy schedule to read and review the draft we submitted.

Reviewer 2 Report

Review about article titled: Serum AMH test for warning of premenopausal endometrial proliferative lesions.

The authors present a good informative retrospective diagnostic study to examine the value of serum AMH levels as a warning indication of premenopausal endometrial proliferative lesions. It was enjoyable to review this clinical study.  I have the following comments:

1.     I suggest modifying the title to something like Serum AMH levels might indicate premenopausal endometrial lesions.

2.     In general, the manuscript is well written, but the abstract needs to be more objective/specific, what was done and how. In the abstract mentioned the number of samples used in each group so the reader gets that information and starts creating the mental map. I kept reading to find that information. 

3.     Line 98, be concise just write Rotterdam criteria was used, we all know those details.

4.     Thin ovulation is not commonly used, just say anovulation. 

5.     Sentence 63-65, this sentence is repeated and using different references, please correct that!

6.     This is a good study because it analyzed a significant number of samples, but the results are applicable only to this specific population. It will be interesting to see the reproducibility in a heterogeneous population. 

7.     The authors correctly addressed the limitations of the study.

8.     This manuscript needs a better discussion of the results and speculations. For example, I was waiting to read more about the difference in PCOS patients and higher BMI and the difference in AMH levels.  They should add that to the discussion.

9.     The BMI and age are narrow, it would be good to see their reasoning and expectations in wide ranges in other populations.

10.  It is a good proof-of-concept that merits further investigation. 

Author Response

Rewiewer2:

The authors present a good informative retrospective diagnostic study to examine the value of serum AMH levels as a warning indication of premenopausal endometrial proliferative lesions. It was enjoyable to review this clinical study.

 I have the following comments:

  1. I suggest modifying the title to something like Serum AMH levels might indicate premenopausal endometrial lesions.
  2. In general, the manuscript is well written, but the abstract needs to be more objective/specific, what was done and how. In the abstract mentioned the number of samples used in each group so the reader gets that information and starts creating the mental map. I kept reading to find that information.
  3. Line 98, be concise just write Rotterdam criteria was used, we all know those details.
  4. Thin ovulation is not commonly used, just say anovulation.
  5. Sentence 63-65, this sentence is repeated and using different references, please correct that!
  6. This is a good study because it analyzed a significant number of samples, but the results are applicable only to this specific population. It will be interesting to see the reproducibility in a heterogeneous population.
  7. The authors correctly addressed the limitations of the study.
  8. This manuscript needs a better discussion of the results and speculations. For example, I was waiting to read more about the difference in PCOS patients and higher BMI and the difference in AMH levels. They should add that to the discussion.
  9. The BMI and age are narrow, it would be good to see their reasoning and expectations in wide ranges in other populations.
  10. It is a good proof-of-concept that merits further investigation.

REPLY

First, thank you very much for your review comments. Regarding your questions, we provide the following feedback:

  1. Thank you very much for your suggestion. Our team has adopted your opinion to discuss postmenopause and has made corresponding modifications in the revised manuscript.
  2. We modified the abstract and explained the number of sample cases.
  3. We have reduced the description of the Rotterdam standard.
  4. Our language for ovulation dysfunction has been unified as Ovulatory dysfunction (OD).
  5. The problem has been corrected.
  6. Due to the utilization of data from a singular medical center and the implementation of stringent inclusion and exclusion criteria, the generalizability of the findings from this study is restricted. In subsequent investigations, our emphasis will be on encompassing a broader population and establishing collaborations with diverse research centers to validate the replicability of our study results.
  7. Thank you for your approval of this article.
  8. Since this paper focuses on the early warning ability of AMH to EPL, the discussion part mainly focuses on the early warning screening of EPL. Our team is currently working on another study that delves into the correlation between AMH and EPL. That paper is currently in the submission stage and focuses on the relationship between serum AMH and EPL, encompassing both Polycystic Ovary Syndrome (PCOS) and non-PCOS cases. The article discussed the potential role of AMH and BMI in the development of EPL. We hope that article is published successfully and provides you with valuable additional insight.
  9. We also realize that the findings of this paper have limited application. In future studies, we will focus on a wider population and try to cooperate with different research centers to verify the reproducibility of the results of this study.
  10. Thank you for your approval of this article.

Thank you again for taking time out of your busy schedule to read and review the draft we submitted.